# Does MIS Play a Role in the Treatment of Advanced Ovarian Cancer?

**DOI:** 10.3390/cancers14153579

**Published:** 2022-07-22

**Authors:** Augusto Pereira, Javier F. Magrina, Paul M. Magtibay, Joao Siufi Neto, Daniela F. S. Siufi, Yu-Hui H. Chang, Tirso Perez-Medina

**Affiliations:** 1Department of Gynecologic Surgery, Puerta de Hierro University Hospital, 28222 Madrid, Spain; tirsoperezmedina@gmail.com; 2Department of Medical and Surgical Gynecology, Mayo Clinic Hospital, Phoenix, AZ 85054, USA; jmagrina@mayo.edu (J.F.M.); magtibay.paul@mayo.edu (P.M.M.); 3Gynecologic Division, BP—A Beneficencia Portuguesa de Sao Paulo, Faculdade de Medicina FMUSP, Universidade de Sao Paulo, Sao Paulo 01323-001, Brazil; joaosiufi@gmail.com (J.S.N.); danielafsantos@me.com (D.F.S.S.); 4Department of Biostatistics, Mayo Clinic, Scottsdale, AZ 85259, USA; chang.yuhui@mayo.edu

**Keywords:** interval cytoreduction, neoadjuvant chemotherapy, ovarian malignancy, primary cytoreduction, survival, progression-free survival

## Abstract

**Simple Summary:**

Minimally invasive surgery can be used for interval debulking after neoadjuvant chemotherapy in selected patients with ovarian cancer initial disease unresectable by laparotomy without compromising survival. The main benefit of minimally invasive surgery for primary and interval debulking is a shorter hospital stay. There are fewer intestinal resections at interval debulking compared with primary debulking.

**Abstract:**

Neoadjuvant chemotherapy allows a minimally invasive approach for interval debulking in patients with ovarian cancer considered unresectable to no residual disease by laparotomy at diagnosis. The aim of the study was to evaluate the type of surgical approach at interval debulking (ID) after three courses of carboplatin and taxol in patients with unresectable ovarian cancer at diagnosis compared with the type of surgical approach at primary debulking (PD). A secondary objective was to compare the perioperative outcomes of MIS vs. laparotomy at ID. A retrospective review of the type of surgical approach at ID following three courses of carboplatin and taxol was compared with the surgical approach at PD, and a review of the perioperative outcomes of MIS vs. open at ID was performed during the period from 21 January 2012, through 21 February 2013, for stage IIIC > 2 cm or IV epithelial ovarian cancer (EOC) unresectable at diagnosis and the surgical approach at PD. During the study period, 127 patients with stage IIIC or IV EOC met the inclusion criteria. Minimally invasive surgery (MIS), laparoscopic or robotic, was used in 21.6% of patients at ID and in 23.3% of patients at PD. At ID, MIS patients had a shorter hospital stay as compared to laparotomy (2 vs. 8 days; *p* < 0.001). At 5 year follow-up, 31.5% of EOC patients were alive (ID MIS: 47.5% vs. ID open: 30%; PD MIS: 41% vs. PD open: 28%), while 24.4% had no evidence of disease (ID MIS: 39% vs. ID open: 19.5%; PD MIS: 32% vs. PD open: 22%). Among living patients, 22% had evidence of disease. Neoadjuvant chemotherapy is a form of chemo-debulking and allows a minimally invasive approach at interval debulking in about one-fifth of the patients, with initial disease deemed unresectable to no residual tumor at initial diagnosis.

## 1. Introduction

Interval debulking (ID) has become an accepted therapeutic approach for patients with advanced epithelial ovarian cancer (EOC) because of similar or lower rates of morbidity and/or mortality and similar survival rates as for primary debulking (PD) [1,2,3,4,5]. In the United States, ID increased from 8.6% to 22.6% between 2004 and 2013 and has been increasingly performed by using minimally invasive surgery (MIS) [6,7,8,9,10].

MIS has been investigated in patients with suspected early ovarian cancer at surgical staging. Nevertheless, the current literature suggests that MIS for ID is a reasonable approach for selected patients with advanced EOC after neoadjuvant chemotherapy (NACT) [7,8,9,10,11,12,13,14].

The recent literature contains only a few studies on minimally invasive approaches, with some limitations such as small case series rather than randomized controlled studies, heterogeneity in patient selection, unavailability of data on the extent of tumor burden after NACT, residual disease at the end of surgery, and long-term follow up, as well as incomplete survival analysis or absence of recurrence rates. 

We aimed to analyze the impact on overall survival (OS) and progression-free survival (PFS) of NACT according to the type of surgical approach at ID in advanced ovarian cancer unamenable to no residual disease (R0) at diagnosis.

## 2. Materials and Methods

This study was approved by the Mayo Clinic Institutional Review Board. Informed consent was obtained from all patients. The operative reports from the electronic health records were reviewed for patients with EOC stage IIIC > 2 cm and IV undergoing ID or PD from 21 January 2006 through 21 February 2013. Selected ID patients had extensive disease not amenable to complete resection even by laparotomy, initial laparoscopic exploration with detailed description of their disease, International Federation of Gynecology and Obstetrics (FIGO) stage IIIC or IV EOC with abdominal disease > 2 cm, and three courses of primary chemotherapy (PC) consisting of carboplatin and taxol before and after ID. Patients who did not have laparoscopic exploration or had an incomplete description of disease, underwent PC for reasons other than unresectable disease, or had different chemotherapy drugs, different number of courses, or different route (IP) before or after ID were excluded. PD patients had initial laparoscopic exploration with detailed description of their disease, as well as International Federation of Gynecology and Obstetrics (FIGO) stage IIIC or IV EOC with abdominal disease > 2 cm.

Decisions for primary debulking or chemotherapy, as well as regarding the type of surgical approach at PD and ID, laparotomy or MIS, were made by two gynecologic oncologists (J.F.M., P.M.M.) equally experienced in laparotomy and MIS for EOC. Decisions were simply on the basis of whether a complete cytoreduction to R0 appeared feasible or not at laparoscopic exploration performed at PD and ID. Patients selected for PC had unresectable disease to R0 even by laparotomy. Patients selected for PD, either laparotomy or MIS, had disease amenable to R0.

Data regarding demographic information, FIGO stage, tumor histologic findings, surgical approach, type of procedures, blood transfusions, complications (wound infection; pulmonary, cardiovascular, or renal complications; deep vein thrombosis; abdominal or pelvic abscess; anastomotic leak; urinary tract infection), length of hospital stay, 30 day mortality, and survival were retrieved from the patients’ electronic health records and the Mayo Clinic Tumor Registry.

### Statistical Analysis

Dichotomous or categorical variables were summarized as numbers and percentages; continuous variables were summarized as means (SD) and medians. Differences in distributions of dichotomous variables were analyzed with Fisher exact tests. Differences between distributions of continuous variables were analyzed with Kruskal–Wallis tests. The primary statistical endpoints were OS and PFS. OS was calculated from the date of the diagnosis to the date of death or last follow-up. PFS was calculated from the date of primary surgery to the diagnosis of recurrence. Kaplan–Meier survival analysis was used to analyze time-to-event data, and the significance of the difference between the curves of the subgroups was assessed with the log-rank test; the results are shown in Figure 1. Statistical software (SAS version 9.3, SAS Institute Inc., Cary, NC, USA) was used for data analysis. A *p*-value ≤ 0.05 was considered statistically significant.

## 3. Results

Demographic characteristics, comorbid conditions, pathologic factors, and FIGO stages of the cohort are shown in Table 1 and Table 2. A group of 30 patients undergoing ID met the inclusion criteria. The only difference in perioperative outcomes for MIS patients undergoing ID was a shorter hospital stay by 6 days. There were no conversions to laparotomy. Results of perioperative outcomes for the PD group and for the ID group are shown in Table 3 and Table 4.

No major differences were noted for the type of procedures performed by MIS vs. laparotomy at PD and ID with the exception of hysterectomy and unilateral or bilateral salpingo-oophorectomy. Results of comparison of procedures for the PD group and for the ID group are shown in Table 5 and Table 6.

At the 5 year follow-up, 40 patients (31.5%) were alive and 87 (68.5%) were dead. The median follow-up for the entire cohort was 31 months (range, 0.50–116 months). After ID, the 5 year OS for MIS patients was 47.5% compared to 30% for laparotomy patients. After PD, the 5 year OS for MIS patients was 41% as compared to 28% for laparotomy patients.

At the 5 year follow-up, a total of 31 patients (24.4%) had no evidence of disease, while 96 (75.6%) had evidence of recurrence. Among the 40 living patients, 31 (78%) were with absence of disease and nine were with evidence of disease (22%). After ID, the 5 year PFS for MIS was 39% compared to 19.5% for laparotomy. After PD, the 5 year PFS was 32% for MIS patients and 22% for laparotomy patients.

There were no differences in the PFS and OS curves for PD and ID by laparotomy and MIS (Figure 1). When s(t) values were compared at different points on the Kaplan–Meier curves, we observed higher OS and PFS favoring MIS over laparotomy. During the period from 25 to 50 months there was a 34% higher OS for ID MIS patients and a 20% higher OS higher for PD MIS patients. Subsequently, during the period between 55 to 60 months, there was a 20% higher PFS for ID MIS and a 10% higher PFS for PD MIS patients.

## 4. Discussion

Because NACT is a form of chemical cytoreduction (primary chemo-debulking) we wanted to evaluate its impact on the surgical approach at ID in unresectable patients at diagnosis. Interestingly, the use of MIS was feasible in 21.6% of patients at ID, similar to our approach at PD (23.3%). Melamed et al. [10] reported using MIS in 15% of patients at ID after primary chemotherapy and with initial widespread disease. Using MIS at ID is a major benefit of primary chemotherapy because these are patients usually with initial widespread disease not amenable to complete resection even by laparotomy or in poor clinical conditions to tolerate a primary debulking.

The results of MIS for primary, interval, and secondary debulking have been reported [7,8,9,10,11,12,13,14,15,16,17,18,19,20]. Gueli Alletti et al. [9] showed the feasibility and safety of MIS for ID in a multicenter trial of patients with a complete clinical response to NACT. However, some authors suggest that MIS should be limited to standard cytoreductive procedures of low complexity [12] or when gastrointestinal, splenectomies, or liver resections are unlikely to be performed [21]. In our study, when we compared procedures for the ID group, liver, splenectomy, or bowel resections were lower in the MIS group, but the difference with the laparotomy group was not statistically significant (Table 6).

The use of MIS at ID should not compromise a complete tumor resection because, as is the case for PD, complete tumor resection at ID is associated with increased survival as compared to incomplete resection [17]. Some authors have described that the extent of tumor burden after NACT remains a significant prognostic factor [21,22].

Comparison studies of MIS and laparotomy for ID have shown as major benefits a shorter hospital stay and reduced blood loss, with similar rates of complications, optimal debulking [7,10,14,23], operative mortality [10], readmissions [10], recurrence [16], and survival [7,10,23]. Furthermore, our data confirm that the main benefit of MIS at ID, as compared to laparotomy, was a shorter hospital stay of 6 days, a finding already noted by Melamed et al. Rates of blood transfusions, complications, and resections to R0 were similar (Table 2).

Additional reported outcomes include a shorter interval to start of chemotherapy after ID [7,16], and comparable or longer operating times for MIS. The reported rates of conversion to laparotomy are low [8,10] and can be minimized with a careful laparoscopic exploration, as reported by Fagotti et al. [24] and Gueli Alletti et al. [9]. In our study, no MIS procedures were converted to laparotomy. Melamed et al. [10] reported the largest retrospective cohort study comparing MIS vs. laparotomy for ID. They used the National Cancer Database and found that, as in our study, the main benefit of MIS was a shorter hospital stay. It remains to be determined if the administration of PC in patients with less extensive disease at initial diagnosis would result in an increased use of MIS at ID with its associated benefits.

Two meta-analyses were published recently [14,25], the first one involving 10 studies of ovarian cancer surgical staging in early-stage EOC (MIS versus open) not demonstrating deleterious survivals or recurrences associated with MIS for ovarian cancer. In relation to cytoreductive surgery for advanced ovarian cancer, we identified four studies that compared survival by MIS at ID with MIS at PD [7,10,11,16]. The PFS was not reported in any of the studies. Only the study by Alletti et al. [7] observed a significantly improved PFS (6 months) in univariate analysis for MIS patients as compared to laparotomy. This difference was not significant in multivariate analysis. In none of the four studies cited above [25] was there an association between surgical approach and mortality. Our data suggest that 75.6% of patients progressed at 5 years, of which only nine patients were alive (22% of the total 5 year survivors).

The second meta-analysis included 38 studies with 8367 women with ovarian cancer. The survival outcomes of EOC patients were similar between the MIS and the laparotomy groups, concluding that EOC patients can safely be operated using MIS [14]. Due to insufficient data, Tang et al. were unable to compare disease-free survival or postoperative recurrence rate between both groups.

In our survival analysis, there were also no significant differences when comparing the different treatment options. It is important to emphasize the OS benefits of MIS in the treatment of advanced ovarian cancer. First, the 5 year overall survival was 17.5% higher for ID MIS patients and 13% higher for PD MIS patients compared to laparotomy. Second, some s(t) values in the Kaplan–Meier curves for OS and PFS between the second and fourth years were improved for MIS patients (Figure 1).

The present study had several limitations resulting from its retrospective nature and small number of patients due to patient selection. Differences existed in the location and spread of disease, which were major factors in the decision for primary debulking vs. chemotherapy and for the surgical approach. Patients amenable to MIS had a lesser extent of disease than those operated by laparotomy.

## 5. Conclusions

NACT allows the use of MIS in patients with initial disease not amenable to complete resection, R0, by laparotomy. The main benefit of MIS at ID observed was a shorter hospital stay. Current evidence suggests that the survival of MIS in selected patients with advanced ovarian cancer is not inferior to that in those operated by laparotomy.

## Figures and Tables

**Figure 1 cancers-14-03579-f001:**
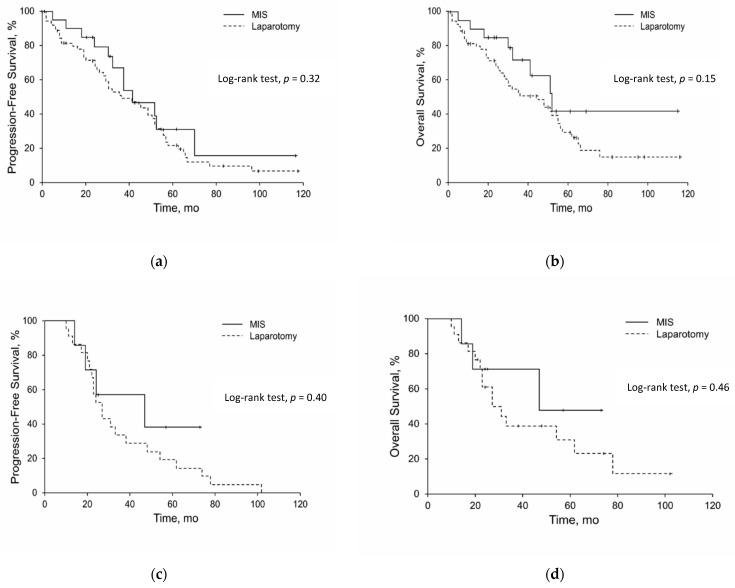
Comparisons of progression-free and overall survival: (**a**) progression-free survival for primary debulking; (**b**) overall survival for primary debulking; (**c**) progression-free survival for interval debulking; (**d**) overall survival for interval debulking. MIS indicates minimally invasive surgery.

**Table 1 cancers-14-03579-t001:** Demographic variables, tumor factors, and FIGO stages for the primary debulking group.

Variable	No. (%) ^a^			
	MIS (*n* = 21)	Laparotomy (*n* = 76)	Total (*n* = 97)	*p*-Value
Age at diagnosis, years				0.31 ^b^
Mean (SD)	64.2 (11.6)	66.7 (9.8)	66.2 (10.2)	
Median	63	67	66	
Body mass index, kg/m^2^				0.92 ^c^
Underweight, <18.5	2 (9.5)	5 (6.6)	7 (7.2)	
Normal weight, 18.5–24.9	9 (42.9)	33 (43.4)	42 (43.3)	
Overweight, 25.0–29.9	7 (33.3)	23 (30.3)	30 (30.9)	
Obese, >30	3 (14.3)	15 (19.7)	18 (18.6)	
Comorbid conditions	13 (61.9)	56 (73.7)	69 (71.1)	0.29 ^c^
Ascites	6 (28.6)	40 (52.6)	46 (47.4)	0.05 ^c^
Pleural effusion	2 (9.5)	13 (17.1)	15 (15.5)	0.40 ^c^
FIGO stage				0.07 ^c^
IIIC	20 (95.2)	59 (77.6)	79 (81.4)	
IV	1 (4.8)	17 (22.4)	18 (18.6)	
Tumor type				0.19 ^c^
Nonserous	2 (9.5)	17 (22.4)	19 (19.6)	
Serous	19 (90.5)	59 (77.6)	78 (80.4)	

FIGO, International Federation of Gynecology and Obstetrics. ^a^ Unless otherwise indicated. ^b^ Wilcoxon rank sum test. ^c^ χ^2^ test.

**Table 2 cancers-14-03579-t002:** Demographic variables, tumor factors, and FIGO stages for the interval debulking group.

Variable	No. (%) ^a^			*p*-Value
	Laparoscopic/Robotic Surgery (*n* = 7)	Laparotomy (*n* = 23)	Total (*n* = 30)	
Age at diagnosis, years				0.90 ^b^
Mean (SD)	67.6 (7.7)	67.2 (10.4)	67.3 (9.7)	
Median	71	69	69.5	
Body mass index, kg/m^2^				0.47 ^c^
Underweight, <18.5	0 (0)	2 (8.7)	2 (6.7)	
Normal weight, 18.5–24.9	1 (14.3)	8 (34.8)	9 (30)	
Overweight, 25.0–29.9	5 (71.4)	12 (52.2)	17 (56.7)	
Obese, >30	1 (14.3)	1 (4.3)	2 (6.7)	
Comorbid conditions	5 (71.4)	18 (78.3)	23 (76.7)	>0.99 ^c^
Ascites	6 (85.7)	17 (73.9)	23 (76.7)	>0.99 ^c^
Pleural effusion	1 (14.3)	2 (8.7)	3 (10)	>0.99 ^c^
FIGO stage				0.15 ^c^
IIIC	7 (100)	16 (69.6)	23 (76.7)	
IV	0 (0)	7 (30.4)	7 (23.3)	
Tumor type				>0.99 ^c^
Nonserous	0 (0)	3 (13)	3 (10)	
Serous	7 (100)	20 (87)	27 (90)	

FIGO, International Federation of Gynecology and Obstetrics. ^a^ Unless otherwise indicated. ^b^ Wilcoxon rank sum test. ^c^ χ^2^ test.

**Table 3 cancers-14-03579-t003:** Perioperative outcomes for the primary debulking group.

Variable	MIS (*n* = 21)	Laparotomy (*n* = 76)	Total (*n* = 97)	*p*-Value
Debulking status, No. (%)				
Complete	19 (90.5)	59 (77.6)	78 (80.4)	0.19 ^a^
Operative time, min				0.004 ^b^
Mean (SD)	277 (107.1)	204.4 (59.8)	220.1 (78)	
Median	241	192	199	
Range	(147–509)	(104–356)	(104–509)	
Hospital stay, days				0.002 ^b^
Mean (SD)	5.6 (3.5)	9.1 (5)	8.4 (4.9)	
Median	5	8	7	
Range	(2–13)	(2–27)	(2–27)	
Blood transfusion, No. (%)	7 (33.3)	41 (54)		0.09 ^b^
Complications, No. (%)				
Intraoperative	1 (4.8)	5 (6.6)	6 (6.2)	0.76 ^a^
Postoperative	5 (23.8)	19 (25)	24 (24.7)	0.91 ^a^
Mortality (30 days), No. (%)	0 (0)	2 (2.6)	2 (2.1)	0.45 ^a^

^a^ Wilcoxon rank sum test. ^b^ χ^2^ test.

**Table 4 cancers-14-03579-t004:** Perioperative outcomes for the interval debulking group.

Variable	Laparoscopic/Robotic Surgery (*n* = 7)	Laparotomy (*n* = 23)	Total (*n* = 30)	*p*-Value
Debulking status, No. (%)				
Complete	7 (100)	20 (87)	27 (90)	0.31 ^a^
Operative time, min				0.30 ^b^
Mean (SD)	286.6 (107.3)	237.2 (75.6)	248.7 (84.7)	
Median	261	243	245.5	
Range	(131–454)	(75–363)	(75–454)	
Hospital stay, days				<0.001 ^b^
Mean (SD)	2.6 (0.8)	7.8 (4)	6.6 (4.2)	
Median	2	8	5	
Range	(2–4)	(3–21)	(2–21)	
Blood transfusion, No. (%)	1 (14.3)	13 (56.5)		0.09 ^b^
Complications, No. (%)				
Intraoperative	0 (0)	1 (4.3)	1 (3.3)	0.57 ^a^
Postoperative	1 (14.3)	5 (21.7)	6 (20)	0.67 ^a^
Mortality (30 days), No. (%)	0 (0)	0 (0)	0 (0)	

^a^ Wilcoxon rank sum test. ^b^ χ^2^ test.

**Table 5 cancers-14-03579-t005:** Comparison of procedures for the primary debulking group.

Procedure	No. (%)			
	MIS (*n* = 21)	Laparotomy (*n* = 76)	Total (*n* = 97)	*p*-Value ^a^
Total hysterectomy with SO	19 (90.5)	47 (61.8)	66 (68)	0.01
SO	2 (9.6)	29 (38.2)	31 (32.0)	0.04
Appendectomy	14 (66.7)	49 (64.5)	63 (64.9)	0.85
Pelvic lymph node dissection	18 (85.7)	59 (77.6)	77 (79.4)	0.42
Aortic lymph node dissection	16 (76.2)	60 (78.9)	76 (78.4)	0.79
Omentectomy	21 (100)	74 (97.4)	95 (97.9)	0.45
Any intestinal resection	6 (28.6)	38 (50)	44 (45.4)	0.08
Diaphragm resection	3 (14.3)	14 (18.4)	17 (17.5)	0.66
Liver resection	0 (0)	3 (3.9)	3 (3.1)	0.36
Splenectomy	0 (0)	6 (7.9)	6 (6.2)	0.18

SO, unilateral or bilateral salpingo-oophorectomy. ^a^ χ^2^ test.

**Table 6 cancers-14-03579-t006:** Comparison of procedures for the interval debulking group.

Procedure	No. (%)			
	MIS (*n* = 7)	Laparotomy (*n* = 23)	Total (*n* = 30)	*p*-Value ^a^
Total hysterectomy with/without SO	6 (85.6)	20 (86.9)	26 (86.6)	0.71
SO	1 (14.3)	3 (13)	4 (13.3)	0.93
Appendectomy	3 (42.9)	12 (52.2)	15 (50)	0.67
Pelvic lymph node dissection	7 (100)	17 (73.9)	24 (80)	0.13
Aortic lymph node dissection	5 (71.4)	16 (69.6)	21 (70)	0.93
Omentectomy	7 (100)	21 (91.3)	28 (93.3)	0.42
Any intestinal resection	0 (0)	6 (26.1)	6 (20)	0.13
Diaphragm resection	2 (28.6)	6 (26.1)	8 (26.7)	0.90
Liver resection	1 (14.3)	4 (17.4)	5 (16.7)	0.85
Splenectomy	0 (0)	4 (17.4)	4 (13.3)	0.24

SO, unilateral or bilateral salpingo-oophorectomy. ^a^ χ^2^ test.

## Data Availability

Data presented in this manuscript are available from the corresponding authors on reasonable request.

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
