# Peer review of "Does MIS Play a Role in the Treatment of Advanced Ovarian Cancer?"

_cancers, 2022, doi:10.3390/cancers14153579_

Round 1

Reviewer 1 Report

In this study, the authors evaluated the type of surgical approach at ID after three courses of carboplatin and taxol in patients with unresectable ovarian cancer at diagnosis compared with the type of surgical approach at PD, and also compared the perioperative outcomes of MIS vs laparotomy at ID. They found that benefit of MIS at ID was a shorter hospital stay, and the survival of MIS patients with advanced ovarian cancer is not inferior to those operated by laparotomy. The data and conclusion are clear and solid and the manuscript is easy to follow.

Author Response

I appreciate the reviewer comments

Reviewer 2 Report

The introduction and Methods need further details to support the study of investigation.

Author Response

Some section of the manuscript has been modified in order to a better comprehension.

Reviewer 3 Report

The role of minimally invasive surgery (MIS) for interval debulking (ID) after neoadjuvant chemotherapy in ovarian cancer is still controversial and not supported by high-quality data. Therefore, the submitted study is of clinical importance and very much to be welcomed. Despite undoubtful strengths, the work would benefit from a careful revision.   Here are my step-by-step suggestions:   Abstract: In the first sentence, a randomly chosen number is used for suitability for MIS. Please, provide a more generalizable information.   Introduction: Line 42/43: The statement that “the results of MIS for ID are well documented [7-10]” is problematic. Firstly, this part of the manuscript should introduce the reader to the topic, so some essential information is required, e.g. what results do the authors mean? Please provide a short and critical summary of relevant studies (3-4 sentences). Second, if the results of MIS for ID were well documented (they are not), the novelty and usefulness of the presented study could be questioned. Line 45/46. The two sentences appear to be unfinished. In addition, more than four studies have been published so far.   Methods: According to which criteria was resectability assessed? A structured scoring, a consensus or just the subjective decision of the experienced surgeon? Did the two qualifying surgeons use identical criteria? Apparently, patients who underwent MIS received far fewer radical resections: no liver resection, no splenectomy, and no bowel resection during ID; similarly much less bowel resection (25 versus 50%) during PDS. Were these steps omitted because they weren't necessary or simply weren't performed? Or was the decision based on preoperative imaging and/or clinical/laboratory parameters? If this is the case, a selection bias is evident and should be taken into account when interpreting the data. Notably, this aspect is not new and was discussed in the work of Matsuo et al. (particularly not referenced in the current manuscript).    Results: The presentation of the results looks a bit chaotic. The authors begin by stating that a group of 30 patients who underwent ID met the inclusion criteria and link this information to Tables 1 and 2. However, Table 1 relates to primary debulking with different patient characteristics. Please be more precise and present the data step by step. Likewise, the accumulation of results in lines 98-108 is not necessary, because it repeats the more understandable presentation in Table 3. Discussion: The original studies used for discussion and comparison are limited to four. However, more than twice as many studies (published 2017-2022) are available but not included. Given the lack of relevant studies, the following papers should be critically discussed when interpreting the presented results.   1) Fagotti A et al. The INTERNATIONAL MISSION study: minimally invasive surgery in ovarian neoplasms after neoadjuvant chemotherapy. Int J Gynecol Cancer. 2019 Jan;29(1):5-9. PMID: 30640676.  2) Matsuo K et al. Minimally invasive interval debulking surgery after neoadjuvant chemotherapy for metastatic ovarian cancer: a national study in the United States. Arch Gynecol Obstet. 2020 Mar;301(3):863-866. PMID: 31980874  3) Morton M et al. Assessing feasibility and perioperative outcomes with minimally invasive surgery compared with laparotomy for interval debulking surgery with hyperthermic intraperitoneal chemotherapy for advanced epithelial ovarian cancer. Gynecol Oncol. 2021 Jan;160(1):45-50. PMID: 33067001  4) Lee YJ et al. Rethinking Radical Surgery in Interval Debulking Surgery for Advanced-Stage Ovarian Cancer Patients Undergoing Neoadjuvant Chemotherapy. J Clin Med. 2020 Apr 24;9(4):1235. PMID: 32344611  5) Tang Q et al. Perioperative and Survival Outcomes of Robotic-Assisted Surgery, Comparison with Laparoscopy and Laparotomy, for Ovarian Cancer: A Network Meta-Analysis. J Oncol. 2022 Apr 30;2022:2084774. PMID: 35535312

Author Response

Abstract: In the first sentence, a randomly chosen number is used for suitability for MIS. Please, provide a more generalizable information.  

RESPONSE: The first sentence of the abstract has been changed

Introduction:

- Line 42/43: The statement that “the results of MIS for ID are well documented [7-10]” is problematic.

Firstly, this part of the manuscript should introduce the reader to the topic, so some essential information is required, e.g. what results do the authors mean? Please provide a short and critical summary of relevant studies (3-4 sentences).

Second, if the results of MIS for ID were well documented (they are not), the novelty and usefulness of the presented study could be questioned.

- Line 45/46. The two sentences appear to be unfinished. In addition, more than four studies have been published so far.  

RESPONSE: The sentences has been added and line 45/46 has been deleted.

Methods: According to which criteria was resectability assessed? A structured scoring, a consensus or just the subjective decision of the experienced surgeon? Did the two qualifying surgeons use identical criteria? Apparently, patients who underwent MIS received far fewer radical resections: no liver resection, no splenectomy, and no bowel resection during ID; similarly much less bowel resection (25 versus 50%) during PDS. Were these steps omitted because they weren't necessary or simply weren't performed? Or was the decision based on preoperative imaging and/or clinical/laboratory parameters? If this is the case, a selection bias is evident and should be taken into account when interpreting the data. Notably, this aspect is not new and was discussed in the work of Matsuo et al. (particularly not referenced in the current manuscript).   

RESPONSE: The resectability criteria was the same for the two surgeons, equally competent in either approach. The decision was simply based on whether or not the entire disease could be removed to no residual tumor by either approach. Properative imaging, physical exam, and lab results provide a resectability estimate. But the final decision to proceed was based on laparoscopic evaluation. The Mastsuo work has been included in the discussion section

Results: The presentation of the results looks a bit chaotic. The authors begin by stating that a group of 30 patients who underwent ID met the inclusion criteria and link this information to Tables 1 and 2. However, Table 1 relates to primary debulking with different patient characteristics. Please be more precise and present the data step by step. Likewise, the accumulation of results in lines 98-108 is not necessary, because it repeats the more understandable presentation in Table 3.

RESPONSE: The result section has been reordered and line 98-108 has been deleted.

Discussion: The original studies used for discussion and comparison are limited to four. However, more than twice as many studies (published 2017-2022) are available but not included. Given the lack of relevant studies, the following papers should be critically discussed when interpreting the presented results.  

1) Fagotti A et al. The INTERNATIONAL MISSION study: minimally invasive surgery in ovarian neoplasms after neoadjuvant chemotherapy. Int J Gynecol Cancer. 2019 Jan;29(1):5-9. PMID: 30640676.  

2) Matsuo K et al. Minimally invasive interval debulking surgery after neoadjuvant chemotherapy for metastatic ovarian cancer: a national study in the United States. Arch Gynecol Obstet. 2020 Mar;301(3):863-866. PMID: 31980874  

3) Morton M et al. Assessing feasibility and perioperative outcomes with minimally invasive surgery compared with laparotomy for interval debulking surgery with hyperthermic intraperitoneal chemotherapy for advanced epithelial ovarian cancer. Gynecol Oncol. 2021 Jan;160(1):45-50. PMID: 33067001  

4) Lee YJ et al. Rethinking Radical Surgery in Interval Debulking Surgery for Advanced-Stage Ovarian Cancer Patients Undergoing Neoadjuvant Chemotherapy. J Clin Med. 2020 Apr 24;9(4):1235. PMID: 32344611  

5) Tang Q et al. Perioperative and Survival Outcomes of Robotic-Assisted Surgery, Comparison with Laparoscopy and Laparotomy, for Ovarian Cancer: A Network Meta-Analysis. J Oncol. 2022 Apr 30;2022:2084774. PMID: 35535312

RESPONSE: The discussion section has been modified based on the five new references.